# The Emerging Relevance of AIM2 in Liver Disease

**DOI:** 10.3390/ijms21186535

**Published:** 2020-09-07

**Authors:** Beatriz Lozano-Ruiz, José M. González-Navajas

**Affiliations:** 1Alicante Institute for Health and Biomedical Research (ISABIAL), 03010 Alicante, Spain; lozano_bea@gva.es; 2Department of Pharmacology, Paediatrics and Organic Chemistry, University Miguel Hernández (UMH), 03550 San Juan, Alicante, Spain; 3Networked Biomedical Research Center for Hepatic and Digestive Diseases (CIBERehd), Institute of Health Carlos III, 28029 Madrid, Spain; 4Institute of Research, Development and Innovation in Healthcare Biotechnology in Elche (IDiBE), University Miguel Hernández, 03202 Elche, Alicante, Spain

**Keywords:** absent in melanoma 2, AIM2, inflammasome, liver disease, NAFLD, NASH, fibrosis, HCC, hepatitis, cirrhosis

## Abstract

Absent in melanoma 2 (AIM2) is a cytosolic receptor that recognizes double-stranded DNA (dsDNA) and triggers the activation of the inflammasome cascade. Activation of the inflammasome results in the maturation of inflammatory cytokines, such as interleukin (IL)-1 β and IL-18, and a form of cell death known as pyroptosis. Owing to the conserved nature of its ligand, AIM2 is important during immune recognition of multiple pathogens. Additionally, AIM2 is also capable of recognizing host DNA during cellular damage or stress, thereby contributing to sterile inflammatory diseases. Inflammation, either in response to pathogens or due to sterile cellular damage, is at the center of the most prevalent and life-threatening liver diseases. Therefore, during the last 15 years, the study of inflammasome activation in the liver has emerged as a new research area in hepatology. Here, we discuss the known functions of AIM2 in the pathogenesis of different hepatic diseases, including non-alcoholic fatty liver disease (NAFLD) and non-alcoholic steatohepatitis (NASH), hepatitis B, liver fibrosis, and hepatocellular carcinoma (HCC).

## 1. Introduction

In addition to fighting pathogens, the immune system is able to respond to signals derived from tissue or cellular damage. Pathogen-derived signals that activate immune cells are named pathogen-associated molecular patterns (PAMPs), while host-derived signals are usually referred to as damage-associated molecular patterns (DAMPs). The recognition of these signals is mediated by several classes of germline-encoded receptors, which are collectively known as pattern recognition receptors (PRRs). Perhaps the best characterized group of PRRs are the toll-like receptors (TLRs), a family of immune activators highly conserved throughout evolution [1]. Other families of PRRs include the nucleotide oligomerization domain (NOD)-like receptors (NLRs), the C-type lectin receptors (CLRs), the RIG-I-like receptors (RLRs), and the absent in melanoma 2 (AIM2)-like receptors (ALRs). Among these families, NLRs and ALRs are the only ones with the ability to induce the assembly and activation of inflammasomes, a molecular platform that activates inflammatory caspases and leads to the secretion of inflammatory cytokines, such as interleukin (IL)-1β and IL-18, and a form of cell death known as pyroptosis.

Inflammation is a key event in the pathogenesis of most liver diseases. Sterile signals from the host and microbial-derived signals from pathogens and the gut microbiota can induce inflammasome activation in different subsets of liver cells, including hepatocytes, kupffer cells, and hepatic stellate cells (HSCs). Then, inflammatory cytokine production and cell death by pyroptosis may have important biological effects on the liver, including the amplification of the inflammatory cascade [2]. Therefore, during the last decade, a considerable research effort has ensued to try to understand the implications of inflammasome activation in almost every liver disease. Most of these efforts have focused on NLR-associated inflammasomes, especially on the NLRP3 (NLR family, pyrin domain containing 3) inflammasome [3,4,5]. NLR inflammasomes have both physiological and pathological roles in obesity and metabolic syndrome, alcoholic liver disease, non-alcoholic fatty liver disease (NAFLD), non-alcoholic steatohepatitis (NASH), hepatitis B virus (HBV) and hepatitis C virus (HCV) infection, drug-induced liver injury, ischemia-reperfusion injury, liver regeneration, liver fibrosis, and hepatocellular carcinoma (HCC) [2,5,6]. In addition, an important role for DNA-sensing inflammasomes, including AIM2 and interferon-γ (IFNγ)-inducible protein 16 (IFI16), has also been reported. In this review, we discuss the importance of AIM2 in the pathogenesis of liver diseases, including NAFLD, HBV infection, fibrosis, cirrhosis, and HCC.

## 2. Brief Overview of AIM2 Inflammasome Activation and Regulation

### 2.1. Mechanism of AIM2 Inflammasome Assembly

Inflammasomes are multiprotein complexes that assemble in the cell cytosol when certain NLRs or ALRs recognize signals associated with infection or tissue damage. With the help, in most cases, of the adaptor protein ASC (apoptosis-associated speck-like protein containing a caspase recruitment domain (CARD)), these receptors recruit and transmit the signal to inflammatory caspases such as caspase-1 and caspase-4 (caspase-11 in mice). Once activated, these caspases mediate the cleavage and maturation of pro-inflammatory cytokines, most notably IL-1β and IL-18, and the protein gasdermin D (GSDMD), which ultimately results in cytokine secretion and cell death by pyroptosis [7,8,9].

AIM2 and other ALRs, such as IFI16, are structurally composed of two main domains; that is, an *N*-terminal pyrin domain (PYD) and a *C*-terminal HIN (hematopoietic expression, interferon-inducible, and nuclear localization) domain. The HIN domain is actually a tandem pair of oligonucleotide/oligosaccharide binding folds [10], which are a group of domains capable of binding to nucleic acids, oligosaccharides, and proteins [11]. The PYD is a member of the death-domain fold family [12] that participates in protein–protein interactions and is commonly found in proteins associated with apoptosis and inflammation [12,13]. Thus, the HIN domain is the part of the AIM2 protein that binds dsDNA in the cytosol, whereas the PYD is the element that relays the signal downstream. The structural mechanism that controls the initial steps of AIM2 activation has been the matter of some debate. One hypothesis states that, in the absence of dsDNA, both HIN and PYD domains interact with each other to keep the protein in an auto-inhibited state [14]. Binding of dsDNA to the HIN domain is sequence-independent, and is mediated by electrostatic interactions between positively charged HIN domain residues and the dsDNA sugar phosphate backbone [14,15]. This binding liberates the PYD from the HIN domain, allowing the PYD to interact with the downstream adaptor. A second hypothesis notes that the role of PYD is not autoinhibitory, and that AIM2 self-assembly is also possible when its cellular concentration is high, driving the initial inflammasome formation without dsDNA [16]. Still, the presence of dsDNA highly reduces the threshold for AIM2 activation; that is, in the presence of dsDNA, AIM2 could assemble even at pico-molar concentrations, but nearly 10,000-fold higher concentration of AIM2 was necessary to self-assemble without dsDNA [16].

The next step in the AIM2 activation sequence is the recruitment of the adaptor protein ASC [17,18] (Figure 1), which is the common adaptor of almost all inflammasomes. This is owing to its bipartite structure, composed of a PYD and a CARD. AIM2 engages ASC through homotypic PYD–PYD interactions [17,18]. The recruitment of multiple ASC molecules results in the irreversible formation of large ASC specks, which involves the oligomerization of the PYD domains into filaments and the cross-linking of these filaments by its own CARD [19]. These specks serve as a signal amplification mechanism for inflammasomes and are a hallmark of their activation [19]. The following event after ASC speck formation is the recruitment of pro-caspase-1 to the complex. Pro-caspase-1 contains a CARD that forms homotypic interactions with the CARD of ASC, incorporating pro-caspase-1 into the ASC filament. Importantly, the CARD of pro-caspase-1 also allows for the oligomerization with other pro-caspase-1 molecules, so multiple molecules are recruited to the ASC speck [20]. Then, pro-caspase-1 molecules auto-cleave themselves at two positions, generating a p20 and a p10 subunit per molecule of pro-caspase-1. Subsequently, two p20 and two p10 subunits form an active heterotetramer, which is the active caspase-1 [20]. Active caspase-1 cleaves potent pro-inflammatory cytokines such as pro-IL-1β and pro-IL-18 into their mature forms and induces their release [21]. Caspase-1 can also cleave the protein GSDMD, removing the *C*-terminal fragment and releasing the *N*-terminal fragment [9]. This *N*-terminal fragment of GSDMD forms large pores in the plasma membrane that result in loss of membrane integrity, release of the mature IL-β and IL-18, and release of other cellular content including DAMPs that amplify the inflammatory signal (Figure 1) [9,13].

### 2.2. AIM2 Inflammasome Regulation

Although AIM2 activation per se does not require transcriptional events, AIM2 expression can be rapidly induced during a priming phase, or signal 1, which is mostly mediated by type I IFN signaling [13,22]. The type I IFN response is typically activated by viral infections, but these cytokines are also induced in response to many bacterial pathogens, mainly through TLR-dependent pathways [23]. Therefore, at least in theory, AIM2 activation can be amplified by multiple pathogen-derived signals. In addition, other mechanisms of AIM2 transcriptional regulation may exist. For example, it was shown that the mRNA expression of most inflammasomes, including AIM2, is affected by the circadian rhythm in mice, showing higher levels in peripheral blood granulocytes during the morning hours and decreasing gradually through the late evening hours [24].

Uncontrolled activation of the inflammasome can lead to chronic inflammation and autoimmune diseases. Therefore, negatively regulating the intensity and duration of the inflammasome response is critical to avoid inflammatory complications. To that end, several intrinsic strategies have evolved. First, several post-translational modifications of ASC have been described, including phosphorylation and ubiquitination [25], which thus have the potential to control not only AIM2, but every ASC-dependent inflammasome as well. Noteworthy, the same type of post-translational modification does not always result in the same outcome. For example, phosphorylation of ASC on Tyr144 (in mice) or Tyr146 and Tyr187 (in humans) is required for NLRP3 and AIM2 activation [26,27], while phosphorylation on the Ser19 and Ser193 residues by IKKα interferes with the cytoplasmic localization of ASC, and thus reduces NLRP3 and AIM2 inflammasome activation [28]. Beyond ASC, AIM2 protein can be directly regulated by tripartite motif 11 (TRIM11) [29]. It was shown that TRIM11 is able to undergo auto-polyubiquitination in the presence of DNA and bind to AIM2 in its ubiquitinated form. In turn, the autophagy cargo receptor p62 recognizes the ubiquitinated TRIM11–AIM2 complex, leading to the delivery of AIM2 to the autophagosome and degradation by selective autophagy [29] (Figure 2).

Another mechanism of inflammasome regulation consists of the existence of decoy proteins that physically interfere with inflammasome assembly or limit ligand availability. Central in this process are the PYD-only proteins (POPs) and CARD-only proteins (COPs). Three POPs have been identified in humans, named POP1–3, plus a truncated POP4 [30]. There are no orthologs for human POPs in mice, but there are at least two predicted POPs in mice [30]. POPs are a family of proteins that contain a PYD, but lack a CARD, and thus are unable to recruit caspase-1. These proteins can bind to the PYD in ASC or the PYD in NLRs or ALRs, thereby blocking PYD–PYD interaction between the inflammasome sensor and the adaptor [31,32]. Among POPs, POP3 has the highest similarity with the PYD of AIM2 and was identified to bind directly to AIM2, competing with ASC and dampening inflammasome activation in mice [33] (Figure 2). COPs work in a similar way, but instead of blocking PYDs, they block CARD–CARD interactions. Three COPs have been described, also known as CARD16, CARD17, and CARD18. All of them are encoded within the caspase-1 locus in the human genome and are absent in the mouse genome [30]. They are highly homologous to the CARD domain of caspase-1, and function by blocking the recruitment of caspase-1 to inflammasomes (Figure 2).

The HIN-200 protein p202 is another decoy protein that can inhibit AIM2 inflammasome in mice [34] (Figure 2). The mouse p202 protein belongs to the ALR family and could be defined as a ‘HIN-only protein’, as it consists of two HIN domains [35], but lacks PYD, and hence is unable to recruit ASC. One HIN domain (HIN-1) interacts with dsDNA, while the other (HIN-2) has affinity for the HIN of AIM2 [35,36]. Therefore, two mechanisms of AIM2 inhibition by p202 have been proposed. First, p202 may compete for DNA with its HIN-1 and limit its availability for AIM2. In addition, this could potentially inhibit other cytosolic DNA sensors. However, p202 has little effect on the cGAS–STING pathway, suggesting that a second mechanism, whereby p202 directly interacts with AIM2, may also exist [13]. This second mechanism was supported by structural studies showing that HIN-2 physically interacts with the HIN of AIM2, preventing ASC clustering and recruitment of caspase-1 [36]. Lastly, AIM2 regulation by a synthetic oligodeoxynucleotide (ODN) has also been described [37]. This ODN, comprised of four repeats of the motif TTAGGG, binds to AIM2 and functions as a competitive inhibitor of the dsDNA–AIM2 interaction, consequently blocking inflammasome assembly [37]. However, the physiological role of ODNs in regulating AIM2 or whether there are similar sequences derived from the host or from pathogens that can efficiently silence AIM2 need further investigation.

## 3. AIM2 in Liver Disease

### 3.1. NAFLD and NASH

The term NAFLD refers to a group of liver conditions, not related to alcohol abuse, in which there is fat accumulation in the liver cells (steatosis) without inflammation. Some patients with NAFLD may progress to a more severe condition called non-alcoholic steatohepatitis (NASH), where fat accumulation is associated with inflammation and liver fibrosis. In turn, NASH can potentially advance to cirrhosis and HCC [38]. The development of NAFLD is highly associated with the metabolic syndrome, an array of metabolic conditions that include obesity, hypertension, hyperlipidemia, and insulin resistance [39,40]. The prevalence of NAFLD is constantly increasing, and currently, almost 25% of adults worldwide have NAFLD [38,41]. From those, at least 10% will develop cirrhosis and/or HCC, thus making NAFLD one of the main causes of end-stage liver disease [41].

The mechanisms responsible for the progression of NAFLD to NASH are still not fully characterized, but inflammasome activation and IL-1β release seem to play an important role [5,42,43,44,45,46]. Nonetheless, few studies have addressed the implication of AIM2 in mouse models of NAFLD or NASH, and human studies are lacking. In a mouse model of methionine-choline deficient (MCD) diet, steatohepatitis was associated with increased AIM2 expression and inflammasome activation [47]. This increase in AIM2 was dependent on activation of TLR signaling through myeloid differentiation factor 88 (MyD88), a common adaptor for most TLRs, in both hematopoietic and non-hematopoietic cells. There was also an increase in high mobility group box 1 (HMGB1) in the liver of MCD-fed mice [47]. HMGB1 is an endogenous danger signal that can activate various members of the TLR family [48]. Therefore, it is possible that the HMGB1–TLR axis is responsible for AIM2 upregulation during fatty liver disease [47]. Similarly, another study found an elevated hepatic AIM2 expression in a model of long-term exposure to a high-fat diet (HFD), which induces NASH in male mice [49]. Together, these studies suggest that AIM2 inflammasome activation might contribute to inflammation and the progression of NAFLD to NASH. On the other hand, AIM2-deficient (*Aim^−/−^*) mice showed spontaneous increase in body weight, fasting glucose, and insulin levels compared with wild type (WT) controls [50]. This was coincident with increased macrophage infiltration and pro-inflammatory markers in the white adipose tissue. However, caspase-1 activity remained unchanged suggesting that these metabolic changes in *Aim2^−/−^*) mice were independent of inflammasome activation [50]. More studies are needed to better comprehend the role of AIM2 in NAFLD and NASH, and to elucidate whether AIM2 could be a therapeutic target in these liver conditions.

### 3.2. HBV Infection

HBV infection is the most common chronic viral infection in the world. It is estimated that around 2 billion people have been infected worldwide, and 350 million are chronic carriers [51]. HBV is an enveloped virus that specifically infects human hepatocytes. It contains a partially double-stranded, relaxed circular DNA (rcDNA) genome that is converted to a covalently closed circular DNA (cccDNA) in the hepatocyte nucleus, which serves as the transcriptional template for the viral replication [51]. Importantly, HBV is not cytopathic, so liver damage is mediated by the immune response against viral antigens. As AIM2 is a dsDNA receptor that initiates an inflammatory cascade, it was soon hypothesized that it could be a contributing factor in HBV-related hepatitis.

AIM2 expression was first reported in peripheral blood mononuclear cells (PBMCs) of patients with acute hepatitis B (AHB) and chronic hepatitis B (CHB). Interestingly, AIM2 expression was higher in patients with AHB than in those with CHB [52]. Moreover, within the group of CHB patients, the highest expression was shown during the immune clearance phase of the disease [52], when the immune system is actively fighting the infection and immune-mediated liver damage occurs [53]. Furthermore, AIM2 expression in PBMCs positively correlated with serum levels of IL-1β and IL-18, and negatively correlated with serum HBV-DNA load and hepatitis B e antigen (HBeAg) [52]. In addition to AIM2, expression of the AIM2-like receptor IFI16 was also upregulated in PBMCs from AHB and CHB patients [54]. In this study, however, AIM2 and IFI16 expression correlated with higher HBV-DNA load, not lower. Therefore, some controversy exists regarding whether AIM2 expression in PBMCs is affected positively or negatively by the presence of HBV-DNA. More importantly, it was reported that HBeAg could inhibit the in vitro activation of AIM2 in PBMCs [54], which may contribute to HBV immunotolerance and persistent infection.

Other studies also evaluated the expression of AIM2 in liver tissue. Han et al. compared the expression of AIM2 protein in liver biopsies from CHB and chronic hepatitis C (CHC) patients, and found that AIM2 is expressed much more frequently in the liver of CHB patients [55]. In liver cells, AIM2 expression was not affected by serum levels of HBeAg, but it positively correlated with viral HBV-DNA load [55]. Moreover, AIM2 expression also correlated with a higher liver inflammation score and with the expression of inflammasome-related markers such as caspase-1, IL-1β, and IL-18. By contrast, there was no association with the degree of liver fibrosis [55]. Similar results were obtained in a different study in which high levels of both AIM2 mRNA and protein were again observed in liver samples from CHB patients, with the highest expression being detected in those patients with more severe liver inflammation [56]. In addition, transfection of human hepatocellular carcinoma cells (HepG2) with the full-length HBV genome triggered an AIM2-dependent production of IL-18 by these cells, showing that the AIM2 inflammasome is functionally active in a human hepatocyte cell line [56].

The implication of AIM2 in extra-hepatic complications of HBV infection has also been explored. AIM2 expression was detected in 81% of renal biopsies from patients with HBV-associated glomerulonephritis, but only in 4% of biopsies from HBV-unrelated glomerulonephritis [57]. AIM2 expression in these samples was not affected by the serum levels of HBeAg, but it positively correlated with HBV-DNA load and the expression of caspase-1 and IL-1β [57]. These data suggest that AIM2 inflammasome activation might contribute to the renal inflammation during HBV infection.

In summary, these reports suggest that AIM2 may have different roles during HBV infection depending on the cell type in which it is expressed. In immune cells, AIM2 may contribute to the immune clearance of HBV. On the other hand, AIM2 expression in hepatocytes could contribute to the inflammatory damage associated with HBV infection.

### 3.3. Liver Fibrosis and Cirrhosis

Liver fibrosis is the result of a wound healing response to chronic live injury that is characterized by excessive deposition of extracellular matrix in the liver. The main causes of liver fibrosis include HBV or HCV infection, alcohol abuse, NASH, and autoimmune hepatitis. Irrespective of the etiology, this repair process is always preceded by inflammation and is largely mediated by innate and adaptive immune mechanisms [58]. A crucial event in this process is the aberrant activation of hepatic stellate cells (HSCs), which differentiate into myofibroblasts that produce type I collagen [59]. HSCs also activate hepatocytes and bone marrow-derived fibroblast to produce collagen and extracellular matrix components [60]. When severe scarring occurs, fibrosis leads to cirrhosis and severe impairment of liver function. This fibrogenic process is slow and, in some cases, it may take decades to progress to cirrhosis, but when cirrhosis is established, several life-threatening complications may appear, such as spontaneous bacterial peritonitis, encephalopathy, and ascites [58,61].

As chronic inflammation is at the core of liver fibrosis and cirrhosis, the role of the inflammasome during liver fibrogenesis has been addressed by several studies [62]. In addition, inflammasome activation may also contribute to the inflammatory complications that occur after cirrhosis is established. However, only a handful of studies have focused on the specific effect of AIM2 in this setting. A study from our group demonstrated that macrophages from the ascitic fluid of cirrhotic patients had elevated expression of caspase-1 and AIM2 compared with blood macrophages from the same patients, whereas the expression of NLRP3, NLRP1, or NLRC4 remained unchanged [63]. Elevated AIM2 expression allowed macrophages from the ascitic fluid to produce large amounts of IL-1β and IL-18 in response to dsDNA, which was much higher than the production observed from blood macrophages from the same patients or from healthy subjects [63]. These data highlight the compartmentalization of the innate immune response in cirrhosis, at least in regards to inflammasome activation. In addition, AIM2 activation correlated with the severity of cirrhosis, because ascitic fluid macrophages from patients with more severe disease produced higher amounts of IL-1β and IL-18 in response to dsDNA [63].

A recent study investigated the role of AIM2 during brucellosis [64], a zoonotic infection that is often associated with liver fibrosis. Using LX-2 cells, a human HSC cell line, the authors show that *Brucella abortus* infection induces IL-1β secretion by HSCs through activation of both NLRP3 and AIM2 inflammasomes. In addition, purified DNA from *B. abortus* was also capable of inducing IL-1β secretion by these cells in an AIM2-dependent manner [64]. Noteworthy, in vivo infection of *Aim2^−/−^*) mice with *B. abortus* resulted in reduced liver fibrosis compared with infected WT mice [64]. Collectively, these results suggest that AIM2 promotes liver fibrosis during brucellosis. Similarly, AIM2 seems to be upregulated during schistosomiasis [65], another parasitic infection strongly associated with liver fibrosis. Schistosomiasis is an inflammatory disease that occurs when eggs from *Schistosoma* species are deposited in the liver, leading to periovular granulomas and fibrosis that can range from scattered to severe portal fibrosis and vascular lesions [66]. AIM2 mRNA and protein expression was increased in the liver of mice infected with *Schistosoma mansoni* [65]. In addition, soluble egg antigens from *S. mansoni* also induced AIM2 mRNA expression in Huh-7 cells [65], a human hepatocarcinoma cell line. However, whether AIM2 activation contributes to liver fibrosis during schistosomiasis still needs to be demonstrated.

Together, these studies touch on the possible role of AIM2 during HSC activation, fibrogenesis, and the inflammatory complications of advanced fibrosis and cirrhosis. However, much more research is needed to better explain the function of AIM2 during liver fibrosis.

### 3.4. Hepatocellular Carcinoma (HCC)

Liver cancer is one of the leading causes of cancer-related deaths worldwide. HCC is by far the most common primary liver cancer, representing around 90% of all cases [67]. Well-known causes of HCC include HCV or HBV infection, alcohol abuse, and NASH. In addition, some contributing factors such as tobacco inhalation and consumption of aflatoxin-contaminated food are also well characterized. HCC is a classic example of an inflammation-associated cancer, as approximately 90% of cases appear after chronic liver inflammation [67,68]. Therefore, the involvement of inflammasome-mediated mechanisms in the pathogenesis of HCC has been the focus of several studies.

AIM2 was originally identified as a tumor suppressor gene in melanoma [69]. Since then, many investigations have reported both pro-tumorigenic and anti-tumorigenic effects of AIM2, which seem to depend on the type of the tumor. Tumor-promoting functions have been reported in squamous cell carcinoma [70], a malignancy that is associated with chronic skin inflammation. On the other hand, AIM2 plays a protective role in breast or intestinal tumors by suppressing proliferation or inducing tumor cell death [71,72,73,74,75]. With regards to HCC, reduced AIM2 expression was observed in HCC tissue compared with distal non-cancerous tissue from the same patients in three independent studies [76,77,78]. In the study by Ma et al. [76], lower AIM2 expression correlated with more advanced HCC, suggesting that loss of AIM2 in cancer cells contributes to HCC progression [76]. Using a xenograft model in nude mice, the authors also show that exogenous overexpression of AIM2 in HCC cells decreased the growth of the transplanted tumors, which was accompanied by a reduction in the mTOR-S6K1 pathway [76]. In vitro, AIM2 expression inhibited proliferation and colony formation of HCC cell lines [76]. In the study by Chen et al. [77], low AIM2 expression was also associated with more malignant features such as poor tumor differentiation, vascular invasion, and lymph node metastasis [77]. Mechanistically, loss of AIM2 expression was associated with epithelial-to-mesenchymal (EMT) transition, cell migration, and metastatic features of HCC cells [77]. This metastatic potential was also assessed in vivo in orthotopic and caudal vein injection mouse models, in which AIM2-silenced HCC cells generated more metastatic nodules in both liver and lung [77]. However, contrary to the study by Ma et al. [76], AIM2 overexpression or silencing in HCC cell lines did not affect cell proliferation [77]. Intriguingly, Sonohara et al. did not find a significant association between AIM2 expression in HCC tissue and overall survival or recurrence-free survival after HCC resection [78].

The impact of AIM2 in the de novo development of HCC was investigated by our group using diethylnitrosamine (DEN) [79], a widely used genotoxic model of HCC [80]. Of note, genetic inactivation of AIM2 or caspase-1/11 protected from HCC development in this model [79]. The lack of AIM2 seemed to be particularly important during the early stages of hepatocarcinogenesis, as hepatocyte damage and expression of inflammatory and proliferative markers were ameliorated in the livers of *Aim2^−/−^* mice 48 h after DEN administration [79]. AIM2 deficiency was also associated with reduced caspase-1 activation and IL-1β production in the liver, indicating that AIM2 contributes to inflammasome activation during DEN-induced liver damage. AIM2 protein and mRNA were highly expressed in Kupffer cells (KCs), a population of liver resident macrophages, which produced large amounts of IL-1β in response to AIM2 stimulation. Noteworthy, IL-1β production was further increased when KCs were isolated from the livers of DEN-treated mice [79], suggesting that carcinogenic liver damage potentiates AIM2 activation in these cells.

Thus, positive and negative effects of AIM2 in HCC have been reported. These seemingly contrasting roles are, however, not exclusive, and they may reflect a different role for AIM2 in different disease stages (established/advanced HCC vs. early stage) or in different cell types (HCC cells vs. nonparenchymal liver immune cells). Nevertheless, the precise contribution of AIM2 during different stages of HCC and in different liver populations is a subject that deserves further attention.

### 3.5. Acute Liver Failure (ALF)

ALF is defined as a severe hepatocellular injury followed by abnormal liver function, altered coagulation, and hepatic encephalopathy in patients without history of chronic liver disease [81]. The course of the disease is usually fast, with severe alterations occurring within 26 weeks of the onset of illness. ALF is a rare, but life-threatening disease, with an incidence that ranges from 1 to 8 cases per million people, but a mortality of up to 30% [82]. The main causes of ALF include acetaminophen (paracetamol) toxicity and other types of drug-induced liver injury, HBV infection or reactivation, liver ischemia, and autoimmune disease.

Inflammation has been associated with both the initial liver damage and with a second phase of injury after toxic hepatocyte damage, for example, in acetaminophen-induced liver injury. However, the role of the inflammasome in ALF is not very well characterized and, in some cases, such as in acetaminophen-induced injury, remains under some debate [83]. Regarding AIM2, one study showed its involvement during Kupffer cell activation in the concanavalin A (ConA) mouse model of ALF [84]. Both AIM2 mRNA and protein were upregulated in Kupffer cells from ConA-treated mice, and AIM2 silencing dampened IL-1β production by ConA-stimulated Kupffer cells in vitro [84]. The authors also showed that microRNA-223 (miR-223), a regulator of several immune pathways including NLRP3 activation [85,86], negatively regulated IL-1β production by interfering with AIM2 activation in these cells [84]. These data suggest that AIM2 activation in Kupffer cells might be important in the inflammatory events during ALF, but studies in other models and human data are lacking. Nonetheless, it is tempting to speculate that uncontrolled release of dsDNA owing to massive hepatocyte damage could propel the inflammatory cascade via AIM2 activation. Thus, future studies on the role of AIM2 might provide insight into the pathophysiology of ALF. Similarly, it would be interesting to address the role of AIM2 during acute-on-chronic liver failure, a different syndrome characterized by acute and severe liver abnormalities in patients with underlying chronic liver disease or cirrhosis.

## 4. Inflammasome-Independent Role of AIM2 in Health and Disease

Although the function of AIM2 as an inflammasome activator is well established and has been the focus of intensive research, AIM2 is also capable of controlling cellular functions independently of the inflammasome. Two papers published in 2015 clearly demonstrate the inflammasome-independent role of AIM2 in protecting against colorectal cancer (CRC) [71,72]. The study by Man et al. demonstrates that, upon dysregulated Wnt/β-catening signaling, AIM2 suppressed the expansion of tumor-initiating intestinal stem cells at the base of the crypts, and genetic loss of AIM2 resulted in increased stem cell activity and exacerbated tumor development [71]. Intriguingly, this could be reverted by transferring intestinal microbiota from healthy wild type (WT) mice, suggesting that both intrinsic and environmental factors contribute to the development of CRC in these mice [71]. However, it is not clear whether both effects of AIM2, inhibition of the intestinal stem cell population and modulation of the gut microbiota, are directly related. The study by Wilson et al. showed that AIM2 is capable of physically interacting with DNA-dependent protein kinase (DNA-PK) and limiting its activation in colon epithelial cells [72]. DNA-PK is a phosphatidylinositol 3-kinase (PI3K)-related family member that induces phosphorylation and activation of the protein kinase B (Akt), which in turn promotes cell survival. Therefore, through this inflammasome-independent mechanism, AIM2 was able to reduce Akt activation and tumor development in a mouse model of CRC [72]. Later, the suppression of the Akt pathway by AIM2 was also observed in a human CRC cell line [87]. Not only in epithelial cells, but also in immune cells, AIM2 seems to have inflammasome-unrelated functions during gastrointestinal disease. A recent report showed that AIM2 indirectly regulates gastric CD8^+^ T cell frequency during gastritis [88]. This effect is mediated by AIM2 expression in B cells, where it suppresses CXCL16 production, reducing the accumulation of CD8^+^ T cells in the inflamed gastric mucosa [88]. Furthermore, inflammasomes-independent functions of AIM2 have also been observed outside the gastrointestinal tract. In cardiomyocytes, AIM2 was shown to interact with STAT1 and inhibit its phosphorylation, which translated into reduced transcription of pro-inflammatory cytokines [89]. In non-small cell lung cancer (NSCLC) cells, AIM2 colocalizes with mitochondria, where it regulates mitofusin 2 expression and mitochondrial fusion [90].

Altogether, these studies provide compelling evidence that the activities of AIM2 go beyond the inflammasome complex, especially in the gut, and encourage the study of inflammasome-independent functions of AIM2 in other cell types and tissues.

## 5. Concluding Remarks

Perhaps the main function of the AIM2 inflammasome is the protection against microbial infection. Thanks to the ubiquitous nature of DNA, AIM2 has the potential to detect almost any type of pathogen. In addition, uncontrolled cellular damage results in the release of self-DNA, which then may be sensed by AIM2 to assemble the inflammasome complex in the absence of infection, contributing to sterile inflammatory conditions. Moreover, inflammasome-independent roles of AIM2 in signaling pathways and biological processes related to cancer and other diseases are also well documented. Thus, it is clear that the biological importance of AIM2 extends beyond pathogen detection.

The study of AIM2 in different liver diseases (Table 1) is supported by the fact that inflammation, either sterile (e.g., NAFLD and NASH) or induced by pathogens (e.g., viral hepatitis), is the origin of most of these diseases. In addition, another important question is the ability of AIM2 to regulate cellular processes independently of the inflammasome. This has been demonstrated mostly in tumor cells in other organs, such as the gut and lung. Noteworthy, some of the reported functions of AIM2 on HCC cells, discussed herein, might be independent of inflammasome activation, but detailed mechanistic studies will help to further confirm this premise. This could be relevant to emphasize the possible duality of AIM2 in HCC and other diseases, by which positive or negative effects may depend on whether AIM2 activation results in inflammasome activation or in inflammasome-independent pathways. In addition, inflammasome-independent effects of AIM2 may also contribute to cellular regulation or dysregulation in other liver conditions as well, but this has not been thoroughly investigated.

It is also conceivable that AIM2 could have positive or negative effects in a particular liver disease, depending on the cell type in which it is activated (immune vs. non-immune cell) or the stage of the disease. Lastly, the contribution of AIM2 regulators such as p202, POP3, or TRIM11 to liver diseases is unknown and awaits specific studies. Therefore, it is foreseeable that a more detailed study of AIM2 function in the liver will provide a better understanding of the molecular mechanisms that contribute to these diseases.

## Figures and Tables

**Figure 1 ijms-21-06535-f001:**
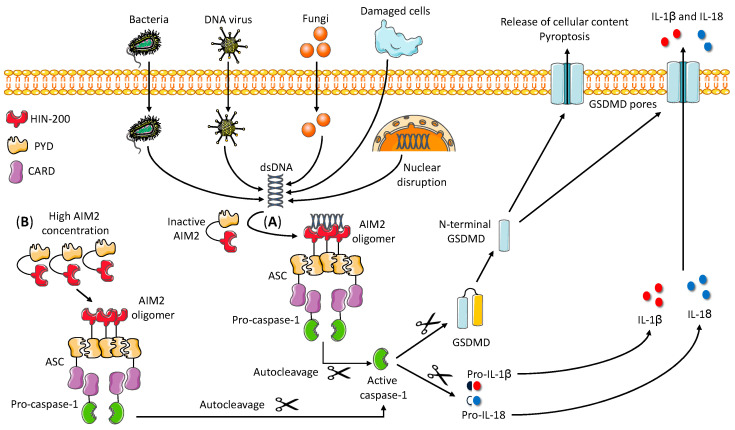
Mechanism of AIM2 inflammasome assembly. Two mechanisms of AIM2 activation have been described. In the presence of dsDNA of any origin, AIM2 loses its auto-inhibitory conformation and the PYD can interact with the PYD of the adaptor ASC (**A**). In addition, a high concentration of AIM2 in the cytosol can also trigger AIM2 oligomerization and ASC recruitment without dsDNA (**B**). In both cases, the next step is the formation of large ASC specks and the recruitment of pro-caspase-1 molecules owing to CARD–CARD interactions. Pro-caspase-1 molecules then undergo auto-cleavage to generate active caspase-1. Active caspase-1 has a dual function: it cleaves pro-IL-1β and pro-IL-18 into the mature IL-1β and IL-18 cytokines, and processes GSDMD, releasing the *N*-terminal fragment. *N*-terminal GSDMD interacts with the plasma membrane and induces the formation of pores, which allows the passive release of IL-1β and IL-18 and also induces pyroptotic cell death and release of cellular contents and alarmins. AIM2—absent in melanoma 2. ASC—apoptosis-associated speck-like protein containing a CARD. CARD—caspase recruitment domain. dsDNA—double-stranded DNA. GSDMD—gasdermin D. HIN—hematopoietic expression, interferon-inducible, and nuclear localization. IL—interleukin. PYD—pyrin domain.

**Figure 2 ijms-21-06535-f002:**
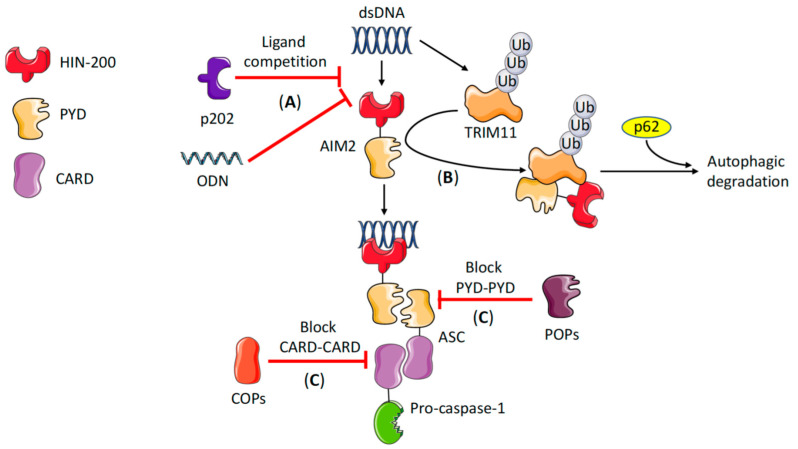
Negative regulation of the AIM2 inflammasome. Several mechanisms have been described with the potential to negatively regulate AIM2. (**A**) The HIN-only protein p202 is present in mouse and is able to directly interact with both the HIN domain of AIM2 as well as with dsDNA, but is unable to recruit ASC; therefore, it can inhibit AIM2 inflammasome activation by blocking the HIN domain of AIM2 and/or by competing for dsDNA. The synthetic ODN TTAGGG can also inhibit AIM2 function by blocking the AIM2–dsDNA interaction. (**B**) TRIM11 is ubiquitinated in the presence of DNA and binds to AIM2. Then, p62 binds to the ubiquitinated TRIM11–AIM2 complex and targets this complex for degradation in the autophagosome, dampening AIM2 inflammasome formation. (**C**) PYD-only proteins (POPs) are decoy proteins that interfere with AIM2 function by blocking PYD–PYD interactions with ASC. Similarly, CARD-only proteins (COPs) are decoy proteins that block CARD–CARD interactions between ASC and pro-caspase-1. AIM2—absent in melanoma 2. ASC—apoptosis-associated speck-like protein containing a CARD. CARD—caspase recruitment and activation domain. dsDNA—double-stranded DNA. HIN—hematopoietic expression, interferon-inducible and nuclear localization. ODN—oligodeoxynucleotide. PYD—pyrin domain. POP—pyrin only protein. COP—CARD-only protein TRIM11—tripartite motif 11. Ub—ubiquitination.

**Table 1 ijms-21-06535-t001:** Reported expression and functions of absent in melanoma 2 (AIM2) in different liver diseases.

Disease	AIM2 Expression and Function	References
NAFLD and NASH	Increased AIM2 expression in MCD and HFD models. Associated with NASH in MCD model	[47,49]
	*Aim2^−/−^* mice show signs of spontaneous metabolic disease	[50]
HBV infection	High AIM2 expression in PBMCs during acute hepatitis B and during the clearance phase of chronic hepatitis B	[52,54]
	Expressed in liver cells of chronic hepatitis B patients. Higher expression correlates with more severe liver inflammation	[55,56]
	HBV genome triggers AIM2 inflammasome activation in HepG2 cell line	[56]
	Expressed in kidney tissue of patients with HBV-associated glomerulonephritis	[57]
Liver fibrosis and cirrhosis	High AIM2 expression and activation in macrophages from ascitic fluid of patients with advanced cirrhosis	[63]
	High AIM2 activation in ascitic fluid macrophages correlates with severity of cirrhosis	[63]
	*B. abortus* triggers AIM2-dependent IL-1β production by human HSCs	[64]
	*Aim2^−/−^* mice develop reduced liver fibrosis after *B. abortus* infection. AIM2 expression is elevated in the liver of mice infected with *S. mansoni*	[64,65]
HCC	Reduced AIM2 expression in HCC tissue compared with non-cancerous liver tissue of the same patients	[76,77,78]
	Lower expression correlates with more advanced HCC features	[76,77]
	DEN-induced liver damage potentiates AIM2 inflammasome activation in KCs	[79]
	*Aim2^−/−^* mice show lower inflammasome activation and reduced HCC development in the DEN model	[79]

NAFLD—non-alcoholic fatty liver disease; NASH—non-alcoholic steatohepatitis; HBV—hepatitis B virus; HCC—hepatocellular carcinoma; MCD—methionine-choline deficient diet; HFD—high fat diet; PBMC—peripheral blood mononuclear cell; IL—interleukin; HSC—hepatic stellate cell; DEN—diethylnitrosamine; KC—Kupffer cell.

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
