# Peer review of "The Emerging Relevance of AIM2 in Liver Disease"

_ijms, 2020, doi:10.3390/ijms21186535_

Round 1

Reviewer 1 Report

In this study, the authors emphasize the emerging role of AIM2, which can recognize cytosolic dsDNA to trigger activation of inflammasome cascade under sterile or pathogenic disease conditions. The manuscript is well-written; however, many mechanistic parts are covered, which are not correlated with liver disease and deviates the manuscript from the title. Please see the comments below:

  1. The introduction section should describe more about the relevance of inflammasome in diseases focusing on liver disease instead of describing the basic mechanisms. Authors can consider all known inflammasomes in the liver disease, and how AIM2 is more relevant out of all?
  2. Sections 2.1 and 2.2 could be removed entirely as these sections centered on molecular mechanisms of AIM2 and not in the context of liver disease. I would encourage to include relevant mechanisms in section 4 'AIM2 in liver disease.'

Author Response

International Journal of Molecular SciencesManuscript ID: ijms-9178761

Point-by-point response -Reviewer 1

We thank the reviewer for his/her suggestions and constructive comments. A point-by-point response to his/her comments is provided below.

In this study, the authors emphasize the emerging role of AIM2, which can recognize cytosolic dsDNA to trigger activation of inflammasome cascade under sterile or pathogenic disease conditions. The manuscript is well-written; however, many mechanistic parts are covered, which are not correlated with liver disease and deviates the manuscript from the title. Please see the comments below:

1.The introduction section should describe more about the relevance of inflammasome in diseases focusing on liver disease instead of describing the basic mechanisms. Authors can consider all known inflammasomes in the liver disease, and how AIM2 is more relevant out of all?

We thank the reviewer for this suggestion. We agree that describing the basic mechanism of inflammasome activation was perhaps more appropriate in a different section of the paper, leaving the introduction more focused on liver disease. So, following the reviewer’s suggestion, we have removed the paragraph with the basic description of inflammasome assembly from the introduction,and wehaveinserted it at the beginning of section 2.1 Mechanism of AIM2 inflammasome assembly(please see lines 76 82of the revised version). We have also expanded the paragraph introducing the importance of the inflammasomes in liver disease (see lines 42 73 of the revised manuscript).

Regarding the relevance of AIM2 in liver diseases in comparison with other inflammasomes,it is not our intention to claimthat AIM2 is the most relevant inflammasome in liver disease. We believe that thedata available in the literatureis notenoughto argue that.There are manystudies reporting important roles forNLRP3and other NLRsin liver disease as well. In fact, most articles reviewing the functionof the inflammasome in liver diseasesfocus mainly on NLR inflammasomes, whilethe role of AIM2 is often overlookedin these reviews.That iswhy we think thatthis review is useful, to focus on the available data regarding the role of AIM2 in liver diseases.

2. Sections 2.1 and 2.2 could be removed entirely as these sections centered on molecular mechanisms of AIM2 and not in the context of liver disease. I would encourage to include relevant mechanisms in section 4 'AIM2 in liver disease.

We understand the point raised by the reviewer. However, we believe that these sections and the associated figures might be useful for the non-specialized reader that is not familiar with the inflammasome field. In addition, one of the comments from Reviewer 2 seems to support these sections and actually asked to introduce some additional information.Therefore, we respectfully decided to maintain these sections in the revised manuscript.

Reviewer 2 Report

Lozano-Ruiz B & González-Navajas deliver in this report a focused review about the role of AIM1 in liver diseases, especially in NASH, Hepatitis B, cirrhosis and HCC. This review is timely and address the importance of AIM2 in such diseases. Because NLRP3 inflammasome is mostly discussed in such context, the AIM2 prism is then original. Overall, the review is very well-written, well-organized and up-to-date. In addition, this review is appropriately illustrated with clear cartoon. Furthermore, authors provide useful tables enabling a quick glimpse at the literature allowing a wide readership to rapidly access to a targeted information.

Finally, we really appreciated the 3rd part demonstrating that AIM2 may have inflammasome-independent manner. Strikingly, from the current limited literature, it seems that such inflammasome-independent activity has only been reported in several diseases/organs but not in liver. This has been superficially discussed by authors in their concluding remarks. Because this review is mainly focused on liver diseases, we were then thinking to put this 3rd part at the end and highlight that such inflammasome-independent activity was mainly reported in other organs/pathophysiological context but not in liver diseases, which may then feed the discussion and allow additional commentaries from authors.

In addition, although we understand that authors mainly focused on chronic liver diseases and hepatitis B, we were wondering whether AIM2 is also involved in Acute Liver Diseases including fulminant hepatitis as demonstrated for other inflammasome including NLRP3. To our knowledge, there is at least one paper discussing that miR-223 controls Kupffer cell activation in an AIM2-dependent manner in a mouse ConA-driven ALF model (Yang et al Cell Physiol Biochem 2014 doi:10.1159/000369658). This might be interesting to add AIM2 regulatory role in ALF too and then provide a complete view of liver diseases.

The part addressing the regulatory pathway of AIM2 is really interesting and thoroughly analyzed as well. However, it mainly focuses on the regulation of the activation step. As author mentioned that a huge quantity of inactivated AIM2 may also trigger inflammasome activation, we were wondering how the initiation step may be regulated as well. In other word, how AIM2 transcriptional expression is regulated? For instance, it has been shown that AIM2 expression is controlled by the circadian machinery (Adamiak M et al Stem cell Rev Rep 2020 DOI: 10.1007/s12015-020-09953-0). It might be interesting to mention such behavior as well.

Finally, in figure 1, authors should remove the red underlining of “Pyroptosis” and page 7 lane 284, authors should edit the typo on hepatocytes.

Author Response

International Journal of Molecular SciencesManuscript ID: ijms-9178761

Point-by-point response Reviewer 2

We thank the reviewer for his/her thorough review and constructive comments. A point-by-point response to his/her comments is provided below.

Lozano-Ruiz B & González-Navajas deliver in this report a focused review about the role of AIM1 in liver diseases, especially in NASH, Hepatitis B, cirrhosis and HCC. This review is timely and address the importance of AIM2 in such diseases. Because NLRP3 inflammasome is mostly discussed in such context, the AIM2 prism is then original. Overall, the review is very well-written, well-organized and up-to-date. In addition, this review is appropriately illustrated with clear cartoon. Furthermore, authors provide useful tables enabling a quick glimpse at the literature allowing a wide readership to rapidly access to a targeted information.

Finally, we really appreciated the 3rdpart demonstrating that AIM2 may have inflammasome-independent manner. Strikingly, from the current limited literature, it seems that such inflammasome-independent activity has only been reported in several diseases/organs but not in liver. This has been superficially discussed by authors in their concluding remarks. Because this review is mainly focused on liver diseases, we were then thinking to put this 3rdpart at the end and highlight that suchinflammasome-independent activity was mainly reported in other organs/pathophysiological context but not in liver diseases, which may then feed the discussion and allow additional commentaries from authors.

We thank the reviewer for this suggestion. It does make sense to change the order of the sections and locate the inflammasome-independent role of AIM2 after the discussion of the different liver diseases. We have changed it accordingly. We have also expanded the discussion on the possible dualityof AIM2(inflammasome vs non-inflammasome effects)in the liver. This change has been included in Section 5 Concluding remarks(see lines449456of the revised manuscript).

In addition, although we understand that authors mainly focused on chronic liver diseases and hepatitis B, we were wondering whether AIM2 is also involved in Acute Liver Diseases including fulminant hepatitis as demonstrated for other inflammasome including NLRP3. To our knowledge, there is at least one paper discussing that miR-223 controls Kupffer cell activation in an AIM2-dependent manner in a mouse ConA-driven ALF model (Yang et al Cell Physiol Biochem 2014 doi:10.1159/000369658). This might be interesting to add AIM2 regulatory role in ALF too and then provide a complete view of liver diseases.

We have followed the reviewer’s suggestion and introduced a new section discussing acute liver failure (ALF). Please see Section 3.5 Acute liver failure (ALF) (please see lines 379  403o f the revised manuscript). Also to our knowledge, the paper by Yang et al. is the only report of AIM2 function during ALF. So, in this new section, we provide a brief definition and overview of ALF for the non-specialized reader and then we discuss the paper by Yang et al.

The part addressing the regulatory pathway of AIM2 is really interesting and thoroughly analyzed as well. However, it mainly focuses on the regulation of the activation step. As author mentioned that a huge quantity of inactivated AIM2may also trigger inflammasome activation, we were wondering how the initiation step may be regulated as well. In other word, how AIM2 transcriptional expression is regulated? For instance, it has been shown that AIM2 expression is controlled by the circadian machinery (Adamiak M et al Stem cell Rev Rep 2020 DOI: 10.1007/s12015-020-09953-0). It might be interesting to mention such behavior as well.

We thank the reviewer for this suggestion as well. We agree that the mechanism that regulates the initiation step of AIM2 activation is an interesting topic.Some inflammasomes such as NLRP3 and AIM2 require a priming signal, also known as signal 1, to promote their expression and the expression of other inflammasome components.It is well known that the expression of AIM2 is controlled by type 1 IFN, however other possible mechanisms that could regulate AIM2 expression are somewhat understudied. The circadian changes in inflammasome receptors, including AIM2, is an interesting observation. Therefore,we have followed the reviewer’s suggestion and introduced a brief discussion on this topic in Section 2.2 Aim2 inflammasome regulation (please see lines 142150of the revised manuscript).

Finally, in figure 1, authors should remove the red underlining of “Pyroptosis” and page 7 lane 284, authors should edit the typo on hepatocytes.

Thanks for identifying these errors. They have been corrected.